# Adsorptive Stripping Voltammetry for Determination of Vanadium: A Review

**DOI:** 10.3390/ma16103646

**Published:** 2023-05-10

**Authors:** Edyta Wlazłowska, Malgorzata Grabarczyk

**Affiliations:** Department of Analytical Chemistry, Institute of Chemical Sciences, Faculty of Chemistry, Maria Curie–Sklodowska University, 20-031 Lublin, Poland; edyta.wlazlowska@onet.pl

**Keywords:** review, vanadium, determination, adsorptive stripping voltammetry, interferences, application

## Abstract

The main purpose of this review is to present methods of adsorptive stripping voltammetry that can be used to determine trace amounts of VO_2_^(+)^ in various types of samples. The detection limits achieved using different working electrodes are presented. The factors influencing the obtained signal, including the selection of the complexing agent and the selection of the working electrode, are shown. For some methods, in order to increase the range of applied concentrations in which vanadium can be detected, a catalytic effect is introduced to adsorptive stripping voltammetry. The influence of the foreign ions and organic matter contained in natural samples on the vanadium signal is analyzed. This paper presents methods of elimination associated with the presence of surfactants in the samples. The methods of adsorptive stripping voltammetry for the simultaneous determination of vanadium with other metal ions are also characterized below. Finally, the practical use of the developed procedures, mainly for the analysis of food and environmental samples, is summarized in a tabular version.

## 1. Introduction

Vanadium is widely present in the earth’s crust, but it is scarce. It is a very valuable metal that is used in industry increasingly more often. It is used in the chemical, glass, ceramics, dyeing and photographic industries, as well as in the manufacturing of nonferrous alloys and high-strength carbon steels. Vanadium enters the environment in large quantities through exhaust gases of coal and petroleum power plants and other industrial enterprises. Vanadium containing vapors that condense to form inhalable aerosols, which cause soil and water contamination and thus also food contamination. In this regard, the main source of vanadium in the environment is human activity, which accounts for almost 67%. Vanadium is an essential element for normal cell growth in small amounts but can be toxic when present in higher concentrations. This silvery gray metal occurs in many oxidation states in the range from −1 to +5, but in natural waters, it occurs mainly as VO^(2+)^ and VO_2_^(+)^. Both species have different nutritional and toxic characteristics. Therefore, the accurate determination of vanadium species in different oxidation states is very important if it is necessary to correctly assess human exposure and associated risks. At the same time, due to the narrow threshold value between the necessary and toxic concentrations for living organisms, the determination of traces of vanadium in various samples is very important and meaningful from the point of view of environmental science and life science [1,2,3].

Many methods for the determination of vanadium have been described in the literature. These include spectrophotometry [4,5,6,7], spectrofluorometry [8], electrothermal atomic absorption spectroscopy (ETAAS) [9,10,11,12], graphite furnace atomic absorption spectroscopy (GFAAS) [13,14,15,16,17,18], neutron activation analysis (NAA) [19], inductively coupled plasma atomic emission spectrometry (ICP–AES) [20,21], inductively coupled plasma mass spectrometry (ICP–MS) [22,23,24] and inductively coupled plasma optical emission spectrometry (ICP–OES) [25,26,27]. The spectrophotometric and spectrofluorimetric methods suffer from low sensitivity values, the NAA method is time consuming and routine analysis of numerous samples is laborious. On the other hand, the ICP and AAS methods, though sensitive, need rigorous sample pretreatment and a high instrumental cost. Of the techniques mentioned above, a review of the literature on the determination of vanadium in environmental samples has been presented [28,29]. In paper [28], a number of important issues with regard to the stability of the vanadium forms and the proposed techniques of their separation were discussed. The separation of the vanadium forms was mainly conducted by extracting it into liquid and solid phases. Combined methods for the analysis of vanadium are also described. They are mainly based on a combination of liquid chromatography and spectroscopic detection. As an example of such a combination, the use of high-performance liquid chromatography with different detection methods was described: ICP–MS [24], FAAS [30] and UV–VIS detection [31]. For the separation of vanadium forms by capillary electrophoresis, UV detection [32,33,34] and chemiluminescence [35] were chosen for quantitative studies of vanadium determination. Analyzing the detection limits presented for the determination of vanadium with the methods presented in [28], the lowest one was obtained by using the ICP–AES method [21] and was equal to 19 ng L^−1^. Wen–yan He et al. reviewed spectrophotometric methods used to determine vanadium [29]. The works collected by them showed that the catalytic determination of vanadium by spectrophotometry can be determined at the level of µg L^−1^, whereas spectrophotometric methods based on redox reactions have detection limits at the level of mg L^−1^. In the literature, many papers have been devoted to the determination of vanadium by GFAAS, which differ mainly in the preparation of the sample for measurement [14,15,16,17,18]. In the work [36], in which the extraction of the cloud point associated with the determination of vanadium was applied, the authors used the method of atomic absorption spectrometry in a high-resolution graphite furnace with a continuous source for the first time. The detection limit obtained was found to be 0.13 µg L^−1^. Detection limits in the µg L^−1^ range can also be obtained by using digital image colorimetry as the detection method [37,38]. In both [37,38], the vanadium determination method was based on extraction. In the case of the research in paper [37], a method based on ultrasonic-assisted microextraction in the liquid phase was developed, while in paper [38], a method based on the microextraction of a single drop in direct immersion was developed.

Much cheaper and simple methods are voltammetric techniques [39,40,41,42,43,44]. The method of choice is adsorptive stripping voltammetry (AdSV), which is also used to determine vanadium in natural samples [45,46,47,48,49,50,51,52,53,54,55,56,57,58,59,60,61,62,63,64,65,66]. The most important parameters and characteristics of these procedures are summarized and presented in Table 1. With the stripping voltammetry technique, the basis of the measurement is the current–potential relationship of the electrode recorded in a three-electrode system, in which the underlying reaction takes place at the working electrode. A schematic diagram illustrating the course of voltammetric measurements is shown in Figure 1.

With the stripping voltammetry technique, the measurement is carried out in two stages. In the first stage, called preconcentration, the component to be determined is accumulated on the electrode; in the second stage, called stripping, the accumulated component undergoes an electrode reaction as a result of a change in the potential of the working electrode. Adsorption stripping voltammetry (AdSV) is one of three voltammetric methods in which, unlike the other two, namely anodic stripping voltammetry and cathodic stripping voltammetry, the accumulation is purely adsorptive in nature and is not related to the course of any Faraday reaction. In the case of the determination of metal ions (M^a+^) by the AdSV method, in the first step, the metal ions react in the solution with the corresponding ligand (L) followed by the adsorption of the complex onto the surface of the working electrode. When the ligand has an inert charge, these processes can be described by Reactions (1) and (2). Once the accumulation step on the electrode surface is complete, it is then polarized in the cathodic or anodic direction, depending on whether the base of the voltammetric curve is a reduction or oxidation reaction. In the case of the reduction of a metal ion, which is by far the more common reaction used in AdSV, the reaction can be written with Equation (3):(1)$$M^{a+}+bL\to \;ML_{b}^{a+}{}_{\left(solution\right)}$$
(2)$$ML_{b}^{a+}{}_{\left(solution\right)}\to ML_{b}^{a+}{}_{\left(electrode\;surface\right)}$$
(3)$$ML_{b}^{a+}{}_{\left(electrode\;surface\right)}+xe^{-}\to M^{a-x}{}_{\left(solution\right)}+bL$$

At the beginning, it is worth emphasizing the advantages of using the AdSV technique. One of the main advantages of this method is certainly the low detection limits obtained. This is possible thanks to the use of an additional stage of accumulation of the determined element on the working electrode. In order to obtain even lower detection limits, it is necessary to use additional catalytic or enzymatic effects. Despite the two stages, the duration of the measurement cycle is short, and typically the duration of the accumulation stage is at most 180 s. An important aspect of performing AdSV measurements is the ability to determine two or more elements during one measurement. The costs of the equipment used are quite low compared to the costs of the equipment used in the other methods mentioned above. Due to the size of the apparatus, it is possible to conduct research in field conditions which, in the event of a threat to a given area, facilitates the speed of the risk assessment and the speed of the reaction to eliminate the contamination. The amount of reagents used is small, because most often there is 10–20 mL of solution in a voltammetric cell. Thanks to so many advantages, this method is very attractive in the case of the trace analyses of many ions [67,68,69,70,71].

In the case of the use of stripping voltammetry in the determination of vanadium, AdSV is mainly used, whereby various complexing agents that form electrochemically active complexes with VO_2_^(+)^ are used and undergo adsorption on various working electrodes. As can be seen in Table 1, dozens of such procedures have been described in the literature, with vanadium detection limits ranging from 2.8 × 10^−12^ to 1 × 10^−7^ mol L^−1^ [45,46,47,48,49,50,51,52,53,54,55,56,57,58,59,60,61,62,63,64,65,66]. In the case of vanadium determination, anodic stripping voltammetry (ASV) is very rarely used, in which the determination is based on redox reactions associated with the change in the oxidation state of vanadium, and few papers on this are available in the literature [41,44]. This is due to the fact that the electrochemical behavior of vanadium is rather complex because of the large number of its oxidation states and its tendency to undergo acid–base reactions, complex formation and polymerization. Vanadium(V) in particular forms many species that are strongly pH dependent. At mercury electrodes, the redox-couple vanadium(V)/(IV) cannot be studied easily in acidic media, because vanadium(V) reacts chemically with mercury whereas the oxidation of vanadium(IV) does not occur within the attainable potential range [44]. Therefore, the voltammetric determination of vanadium is carried out mainly by using the AdSV method. In addition, the AdSV method allows for better sensitivity of determinations, which makes it the first-choice method compared to ASV when we want to obtain a low limit of detection.

This work is dedicated to reviewing the AdSV methods for the determination of vanadium described in the literature, with particular emphasis on the use of catalytic effects, the complexing agents used, the types of working electrodes, the detection limits obtained and their application. The obtained reproducibility of the results and the high sensitivity of the AdSV method are influenced by parameters such as the selection of an appropriate complexing agent, the selection of its concentration and adsorption conditions through the selection of the basic electrolyte, including its type and pH. However, they are also influenced by the parameters in which the measurement is carried out, i.e., the potential and time of accumulation as well as the signal-recording parameters. It is these parameters that have a decisive impact on the attractiveness and possibilities of these procedures, as shown below.

**Table 1 materials-16-03646-t001:** Analytical performance of procedures for the determination of VO_2_^(+)^ by the AdSV method. Procedures were ranked according to increasing LOD.

Working Electrode	Complexing Agent	Catalytic System	LOD (mol L^−1^)	Accumulation Time (s)	Linear Range (mol L^−1^)	Investigated Interferents	Ref.
GCE/PbF	Cupferron	–	2.8 × 10^−12^	15	1 × 10^−11^–2 × 10^−10^	Ca(II), Cd(II), Cu(II), Mg(II), Ni(II), Bi(III) and Fe(III), Triton X–100	[48]
HMDE	Cupferron	VO_2_^(+)^–cupferron–BrO_3_^−^	4.9 × 10^−12^	15	–	Ag(I), TI(I), Cd(II), Cu(II), Co(II), Mn(II), Ni(II), Pb(II), Zn(II), Ti(IV), Bi(III), Cr(III), Hg(II), Te(VI), As(III), Ga(III), Fe(III), Sb(III), Ge(IV), Sn(IV), Pt(IV), Zr(IV), UO_2_^(2+)^ and MoO_2_^(2+)^	[45]
HMDE	Quercetin−5−sulfonic acid	VO_2_^(+)^–QSA–BrO_3_^−^	4.5 × 10^−12^	30	no data–7 × 10^−9^	Na(I), K(I), Ag(I), Be(II), Ca(II), Co(II), Cu(II), Ni(II), Pb(II), Zn(II), Cr(III), Al(III), Sb(III), Zr(IV), Ti(IV), As(V) and MoO_2_^(2+)^	[62]
HMDE	Chloranilic acid	VO_2_^(+)^–chloranilic acid–BrO_3_^−^	9.0 × 10^−12^	100	2 × 10^−10^–5 × 10^−8^	Cu(II), Cd(II), Pb(II), Sn(II), Bi(III), Fe(III), Sb(III), Se(IV), Sn(IV), Te(IV), UO_2_^(2+)^ and MoO_2_^(2+)^, Triton X–100	[52]
Hg(Ag)FE	Chloranilic acid		1 × 10^−11^	90	2.5 × 10^−10^–1 × 10^−7^	Cd(II), Cu(II), Mn(II), Pb(II), Sn(II), Zn(II), Bi(III), Fe(III), Se(IV), UO_2_^(2+)^ and WO_2_^(2+)^, Triton X–100 and HA	[53]
HMDE	2,3–dihydroxynapthlhalene	VO_2_^(+)^–DHN–BrO_3_^−^	1.5 × 10^−11^ 4 × 10^−12^	60 600	5 × 10^−11^–4 × 10^−9^	Cd(II), Co(II), Cu(II), Fe(II), Mn(II), Ni(II), Pb(II), Sn(II), Zn(II), Al(III), As(III), Bi(III), Cr(III), Fe(III), Se(IV), As(V), Cr(VI), MoO_2_^(2+)^ and Se(VI), SDS, DTAC and Triton X–100	[63]
MWEs	Gallic acid	VO_2_^(+)^–GA–BrO_3_^−^	1.7 × 10^−11^	120	1 × 10^−10^–2 × 10^−8^	Ni(II), Sn(II), Fe(III), Al(III), Se(IV), As(V), Cr(VI) and MoO_2_^(2+)^, Triton X–100	[60]
MFE	Cupferron	VO_2_^(+)^–cupferron–BrO_3_^−^	1.6 × 10^−10^	90	2 × 10^−9^–6.9 × 10^−10^	Cd(II), Cu(II), Fe(II), Mn(II), Ni(II), Pb(II), Sr(II), Zn(II), Fe(III), Ti(IV), WO_2_^(2+)^ and MoO_2_^(2+)^	[46]
HMDE	2,3–dihydroxybenzaldehyde	VO_2_^(+)^–2,3–DHBA–BrO_3_^−^	2 × 10^−10^	30	5 × 10^−10^–5 × 10^−8^	Cd(II), Co(II), Cu(II), Fe(II), Mn(II), Ni(II), Pb(II) and Zn(II)	[59]
HMDE	Cupferron	–	2.0 × 10^−10^	50	2 × 10^−9^–2 × 10^−6^	Ag(I), K(I), Li(I), Na(I), Ca(II), Cd(II), Co(II), Cu(II), Mg(II), Mn(II), Ni(II), Ba(II), Zn(II), Al(III), Cr(III), La(III), Fe(III) and Sb(III), Triton X–100	[50]
BiFµE	Cupferron	–	2.5 × 10^−10^	60	8 × 10^−10^–1 × 10^−7^	Tl(I), Cd(II), Co(II), Cu(II), Hg(II), Mn(II), Sn(II), Zn(II), Ni(II), Pb(II), Au(III), Bi(III), Cr(III), Fe(III), Ga(III), Sb(III), In(III), Pt(IV), Ti(IV), WO_2_^(2+)^ and MoO_2_^(2+)^, Triton X–100 and HA	[47]
MFE	Pyrogallol	–	3 × 10^−10^	120	nd–1.5 × 10^−6^	Cd(II), Co(II), Cu(II), Mg(II), Ni(II), Pb(II), Sn(II), Zn(II), Al(III), Cr(III), Fe(III) and As(V), DTAC, SDS and Triton X–100	[61]
HMDE	DMG + catechol	–	3 × 10^−10^	900	nd–1 × 10^−7^	–	[65]
PbFE	Cupferron	–	3.2 × 10^−10^	30	1 × 10^−9^–7 × 10^−8^	Cu(II), Mn(II), Ni(II), Au(III), Bi(III), Ga(III), In(III), Sb(III), Cr(III), Fe(III), Ge(IV), Pt(IV), Ti(IV) and MoO_2_^(2+)^. Triton X–100, SDS, CTAB and Rhamnolipids	[49]
HMDE	Catechol	VO_2_^(+)^–catechol–BrO_3_^−^	6 × 10^−10^ 7 × 10^−11^	15 120	–	Cd(II), Co(II), Cu(II), Mn(II), Ni(II), Zn(II), Al(III), As(III), Fe(III), In(III), Sb(III), Se(IV), Ti(IV), Cr(VI), MoO_2_^(2+)^ and UO_2_^(2+)^, Sodium salt, DBS, Hyamine 1622 and Triton X–100	[55]
ABPE	Alizarin violet	–	6 × 10^−10^	90	8 × 10^−10^–1 × 10^−7^	Ag(I), Ba(II), Be(II), Ca(II), Cd(II), Co(II), Cu(II), Mg(II), Mn(II), Ni(II), Pb(II), Zn(II), Al(III), Bi(III), Cr(III), Fe(III), Ga(III), In(III), Sb(III), Sc(III), Se(IV), Th(IV), Ti(IV), Sn(IV), Zr(IV), WO_2_^(2+)^ and MoO_2_^(2+)^, Triton X–100	[57]
MME	cupferron–oxalic acid–1,3–diphenylguanidine	–	9 × 10^−10^	30	3.79 × 10^−9^–2.84 × 10^−7^	Tl(I), Ag(I), Ba(II), Cd(II), Co(II), Cu(II), Mn(II), Ni(II), Pb(II), Sr(II), Zn(II), Al(III), As(III), Bi(III), Cr(III), Fe(III), Se(IV), Ti(IV), Te(IV), Se(IV), As(V), Nb(V), Cr(VI), MoO_2_^(2+)^, Se(VI), UO_2_^(2+)^ and WO_2_^(2+)^	[64]
HMDE	DMG + catechol	–	2.52 × 10^−9^	900	>3 × 10^−7^	–	[64]
HMDE	Chloranilic acid	–	2.7 × 10^−9^	15	4.59 × 10^−8^–2.9 × 10^−7^	–	[54]
BiFE	Chloranilic acid	VO_2_^(+)^–chloranilic acid–BrO_3_^−^	3.9 × 10^−9^	600	9.8 × 10^−8^–5 × 10^−7^	Ag(I), Au(I), Be(II), Cd(II), Cu(II), Ni(II), Pb(II), Pt(II), Hg(II), Ti(II), Zn(II), Cr(III), Fe(III), Al(III), Sb(III), MoO_2_^(2+)^ and UO_2_^(2+)^	[51]
HMDE	Pyrocatechol violet	–	1 × 10^−8^	180	1 × 10^−8^–6 × 10^−7^	Ag(I), Tl(I), Ca(II), Co(II), Cu(II), Hg(II), Mn(II), Ni(II), Sn(II), Zn(II), Pd(II), Pb(II), Al(III), Fe(III), Rh(III), Ir(IV), Ti(IV), Os(IV), Ce(IV) and Pt(IV), Triton X–100	[58]
HMDE	Chromoxane cyanine R	–	1 × 10^−7^	180	3 × 10^−7^–2.4 × 10^−5^	Ag(I), Cd(II), Ca(II), Ba(II), Mg(II), Zn(II), Co(II), Ni(II), Fe(II), Mn(II), Sr(II), Hg(II), Cr(III), Ga(III), In(III), La(III), Fe(III), Eu(III), Hf(III), Al(III) and WO_2_^(2+)^, SDS and Triton X–100	[56]

HMDE—hanging drop mercury electrode, MFE—mercury film electrode, BiFµE—solid bismuth microelectrode, GCE/PbF—lead-coated glassy carbon electrode, PbFE—lead film electrode, BiFE—bismuth film electrode, Hg(Ag)FE—renewable mercury film silver-based electrode, ABPE—acetylene black paste electrode, MWEs—mercury-coated gold microwire electrodes, MME—mercury multimode electrode.

## 2. Adsorptive Stripping Voltammetry of Vanadium

### 2.1. Procedures Based on Complexation Reactions

Despite the fact that the stripping voltammetry method is one of the most sensitive analytical methods, due to the step of accumulation on the electrode included in the procedure, there are still efforts to further reduce the detection limits of metal ions and hence so many papers are devoted to this issue. A very important element in AdSV is the selection of the complexing agent. It is the complexing agent that is one of the factors affecting the signal of vanadium determined. Its use causes the formation of a complex with a marked ion, which is adsorbed on the surface of the electrode. The complexing agents used in the voltammetric analysis of vanadium are listed in Table 1, together with the detection limits obtained. The most commonly chosen complexing agent in voltammetric procedures for the determination of vanadium is cupferron [45,46,47,48,49,50]. Other complexing agents are also used, such as chloranilic acid [51,52,53,54], catechol [55], chromoxane cyanine R [56], alizarin violet [57], pyrocatechol [58], 2,3–dihydroxybenzaldehyde [59], galic acid [60], pyrogallol [61], quertic–5–sulfonic [62] and 2,3–dihydroxynaphthalene [63]. A mixture of ligands was also used as a complexing agent in the paper [64], namely, cupferron, oxalic acid and 1,3–diphenylguanidine, while in the papers [65,66], a mixture of ligands was used, namely dimethylglyoxime and catechol. Table 2 shows the chemical structures of the most commonly used complexing agents in the procedure for the determination of vanadium by ADSV.

In the case of cupferron concentration optimization, it has been observed that initial additions of cupferron to the sample solution result in a rapid increase in the vanadium signal, while higher cupferron concentrations result in a slower increase [45,48]. Higher concentrations of the complexing agent may also cause adsorption of the free ligand instead of the formed complex, which blocks the surfaces of the working electrode [51]. In the papers [45,46], the concentration of cupferron was optimized in the range from 5 × 10^−6^ mol L^−1^ to 4 × 10^−5^ mol L^−1^, with a concentration of 2 × 10^−5^ mol L^−1^ used for further research. It turned out that the concentration selected as the most optimal in the works [47,48] was 7 × 10^−5^ mol L^−1^. The lowest concentration of cupferron 8 × 10^−5^ mol L^−1^ was used in the work [49], while the highest concentration of 1 × 10^−3^ mol L^−1^ was selected in the vanadium determination procedure using BiµFE as the working electrode [47]. There was also a negative effect of a high concentration of cupferron which interferes with the measurement. Greenway found that any concentration of cupferron higher than 2 × 10^−5^ mol L^−1^ caused a shift in the mercury dissolution potential that interfered with the measurement more than the adjacent oxygen peak [46]. In the case of chloranilic acid (CAA), the selection of its concentration is also very important. When too high a concentration of CAA was used, above 2.5 × 10^−6^ mol L^−1^, the vanadium peak shifted toward more negative potentials and caused an increase in the background current [53]. Using higher concentrations of a complexing agent, such as alizarin violet (AV), with concentrations higher than 2 × 10^−5^ mol L^−1^ also covers the working electrode with the free ligand. This effect inhibits the buildup of the VO_2_^(+)^–AV complex, which results in a drop in the signal [57].

In the work [62] of Rojas–Ramos et al., they compared the influence of two complexing agents on the determination of vanadium(V). This comparison concerned the determination of VO_2_^(+)^ in the presence of quercetin (Q) and quertic–5–sulfonic acid (QSA) using an HMDE as the working electrode. In its structure, QSA contains a sulfone group, which thus increases its polarity. Compounds with a higher polarity dissolve more easily in polar solvents and thus are more difficult to adsorb to the surface of the HMDE electrode. The authors studied the determination of vanadium in the presence of these two ligands under the same conditions. Under appropriately selected conditions, measurements were carried out, based on which it can be clearly stated that QSA is a more suitable ligand than Q. The limit of detection for QSA was 4.5 × 10^−12^ mol L^−1^, while for Q it was 4.9 × 10^−11^ mol L^−1^, and therefore the QSA method is more sensitive than in the case of Q [62].

### 2.2. Catalytic Systems

Catalytic adsorptive stripping voltammetry (CAdSV), which combines the adsorptive preconcentration of the electroactive species with the catalytic response, presents a vast enhancement of the voltammetric response and therefore a sizable lower detection limit in addition to the development of the selectivity of the determination. Voltammetric catalytic methods require the use of a reagent that will selectively amplify the analyte signal. This role is played by compounds causing oxidation of the reduced form of the analyte, thanks to which the same ion can participate in the electrode process many times, which thus results in an increase in the recorded current signals. For any voltammetric technique, the catalytic faradaic current increase can be expressed by the equation below [78]:I_f_ = [adsorptive enrichment] × [catalytic enhancement](4)

Increasing the sensitivity of the determination is the reason both for the use of preconcentration and the introduction of a catalytic effect. The catalytic effect extends the range of concentrations at which measurements can be made. An increase in the ratio of the faradic current of the target ion to the faradic currents of other elements present in the solution will improve the selectivity. In this chapter, various catalytic effects used for vanadium determinations will be presented, and catalytic adsorptive voltammetry stripping will be characterized. Particular attention is paid to the effect of lowering the limit of detection using catalytic reactions.

#### 2.2.1. Catalytic System VO_2_^(+)^–Cupferron–BrO_3_^−^

The catalytic reaction between a complex of vanadium and cupferron was first studied by Wang et al. They proposed a method based on the catalytic reaction of an adsorbed vanadium–cupferron complex in the presence of bromate, which is often considered the best choice in the determination of vanadium. The measurements were carried out in a three-electrode system, in which the working electrode was a hanging drop mercury electrode (HMDE). The cyclic catalytic reaction occurs according to the following scheme:(5)$${\mathrm{VO}}_{2}{}^{\left(+\right)}{\mathrm{Cupf}}_{\mathrm{soln}}\overset{\mathrm{accumulation}}{\to }{\mathrm{VO}}_{2}{}^{\left(+\right)}{\mathrm{Cupf}}_{\mathrm{surf}}$$
(6)$${\mathrm{VO}}_{2}{}^{\left(+\right)}{\mathrm{Cupf}}_{\mathrm{surf}}+e^{-}\to {\mathrm{VO}}^{\left(2+\right)}{\mathrm{Cupf}}_{\mathrm{surf}}$$
(7)$${\mathrm{VO}}^{\left(2+\right)}{\mathrm{Cupf}}_{\mathrm{surf}}+{\mathrm{BrO}}_{3}^{-}+6H^{+}\to {\mathrm{VO}}_{2}{}^{\left(+\right)}{\mathrm{Cupf}}_{\mathrm{surf}}+{\mathrm{Br}}^{-}+3H_{2}O$$

Using this procedure for the determination of vanadium, the authors obtained a reduction in the detection limit by 2–3 orders of magnitude compared to the detection limit obtained using the catalytic–polarographic measurements. Unfortunately, studies have shown the interfering effect of surfactants, which requires the proper preparation of real samples before measurement, for example by irradiating the sample with UV light. Additionally, it is necessary to remove dissolved oxygen, because if it is not removed, a large oxygen peak appears superimposed on the vanadium peak. Unfortunately, all these steps involved in proper sample preparation increase the total analysis time [45].

Another example of the use of cupferron as a complexing agent and bromate ions as an oxidant was proposed by Greenway et al. [46]. The difference between the vanadium determination procedures in the case of Wang and Greenway was the use of a different working electrode. A mercury film electrode (MFE) was used as the working electrode instead of an HMDE. The use of the MFE electrode enabled a flow system to be used, whereas in the case of the HMDE electrode, it was difficult to use a flow system due to the risk of breaking the drop during the flow of the solution. The advantage of the developed method using the flow system was the reduction in the interfering impact of the matrix, and better reproducibility was obtained. Unfortunately, when comparing the obtained limits of detection, we can conclude that a lower limit of detection by two orders of magnitude is obtained when the HMDE is used than in the case of the MFE [46].

#### 2.2.2. Catalytic System VO_2_^(+)^–Chloranilic Acid–BrO_3_^−^

Bobrowski et al. investigated the electrochemical properties of the V–CAA–BrO_3_^−^ catalytic system using the CAdSV and differential pulsed polarography (DPP) techniques [52]. They found that the V–CAA complex adsorbed on the HMDE surface catalytically reacted with BrO_3_^−^ ions, which resulted in an increased signal. When the vanadium signal is recorded using the CAdSV technique, the peak with the addition of BrO_3_^−^ is 275 times higher than without bromate ions in the solution. The use of chloranilic acid allows for the determination of VO_2_^(+)^ over a very wide range of concentrations. Higher vanadium concentrations are determined by DPP, in which the V–CAA complex gives well-defined peaks proportional to the concentration of VO_2_^(+)^. The highest sensitivity was obtained by using the AdSV method, whereby the use of the V–CAA–BrO_3_^−^ catalytic system in combination with the adsorptive accumulation of the V–CAA complex led to a large increase in the voltamperometric response and thus sensitivity for the determination of VO_2_^(+)^ [52].

#### 2.2.3. Catalytic System VO_2_^(+)^–Catechol–BrO_3_^−^

Cyclic adsorptive stripping voltammetry for the determination of vanadium using catechol as a complexing agent was proposed by Vege and Berg [55]. Unfortunately, preliminary research showed the lack of stability of the VO_2_^(+)^ signal in the presence of catechol and bromate ions introduced to increase the sensitivity of the assays through a catalytic effect. This was due to the slow and gradual oxidation of catechol by bromate ions. In order to eliminate the unstable signal, the authors conducted research in the following stages:Heat a mixture of catechol and bromate ion solutions in a microwave oven, which resulted in the formation of orthobenzoquinone, i.e., the oxidized form of catechol.Reduction of orthobenzoquinone to catechol and the formation of a complex with vanadium(V), which is the stage of adsorption.In the range of potentials −0.8 V–0.1 V, the vanadium(V)–catechol complex is reduced to vanadium(IV)–catechol.The vanadium(IV)–catechol complex is chemically oxidized to the vanadium(V) form by bromate ions, where the vanadium(V)–catechol complex is ready for repeated reduction, thanks to which the catalytic increase in the reduction current is visible [55].

#### 2.2.4. Catalytic system VO_2_^(+)^–2,3–DHBA–BrO_3_^−^

In their work, Pover et al. investigated the influence of BrO_3_^−^ ions on the peak of the VO_2_^(+)^–2,3–dihydroxybenzaldehyde (2,3–DHBA) complex [59]. The authors concluded that in this case, an oxidation–reduction reaction occurs between the vanadium and 2,3–DHBA, which results in complexes VO^(2+)^ tris–chelate that are reduced to V(III) complexes. The catalyst for the reduction of bromate ions is the formed complex VO^(2+)^ tris–chelate. The adsorptive–catalytic reaction occurs according to the following scheme:(8)$${\mathrm{VO}}_{2}{}^{\left(+\right)}+2,3-\mathrm{DHBA}\to {\mathrm{VO}}^{\left(2+\right)}+\mathrm{oxidation}\;\mathrm{products}\;\mathrm{of}\;2,3-\mathrm{DHBA}$$
(9)$${\mathrm{VO}}^{\left(2+\right)}+3\left(2,3-\mathrm{DHBA}\right)\to {\left[{\mathrm{VO}}^{\left(2+\right)}\left(2,3-\mathrm{DHBA}\right)3\right]}^{2-}↔{\left[{\mathrm{VO}}^{\left(2+\right)}{\left(2,3-\mathrm{DHBA}\right)}_{3}\right]}_{\mathrm{ads}}^{2-}$$
(10)$${\left[{\mathrm{VO}}^{\left(2+\right)}{\left(2,3-\mathrm{DHBA}\right)}_{3}\right]}_{\mathrm{ads}}^{2-}+e\;↔{\left[V\left(\mathrm{III}\right){\left(2,3-\mathrm{DHBA}\right)}_{3}\right]}^{3-}\;$$
(11)$${\left[V\left(\mathrm{III}\right){\left(2,3-\mathrm{DHBA}\right)}_{3}\right]}^{3-}+{\mathrm{BrO}}_{3}^{-}\to \left\{{\left[V\left(\mathrm{III}\right){\left(2,3-\mathrm{DHBA}\right)}_{3}\right]}^{3-}{*\;\mathrm{BrO}}_{3}^{-}\right\}$$
(12)$$\left\{{\left[V\left(\mathrm{III}\right){\left(2,3-\mathrm{DHBA}\right)}_{3}\right]}^{3-}{*\;\mathrm{BrO}}_{3}^{-}\right\}+2H^{+}\to {\left[{\mathrm{VO}}^{\left(2+\right)}{\left(2,3-\mathrm{DHBA}\right)}_{3}\right]}^{2-}+{\mathrm{BrO}}_{2}+H_{2}O$$

In Equation (8), a polarographically active complex (PAC) is formed, which is a limitation of the process because once formed in the solution, the PAC adsorbs on the electrode surface and takes up space for the VO_2_^(+)^–2,3–DHBA complexes [59].

#### 2.2.5. Catalytic System VO_2_^(+)^–GA–BrO_3_^−^

Another catalytic system used to increase the sensitivity of vanadium determinations using gallic acid (GA) was VO_2_^(+)^–GA–BrO_3_^−^ [60]. In this case, mercury-coated gold microwire electrodes (MWEs) were used as the working electrode. The parameters optimized for the adsorption of complexes V (V)–GA at the MWEs were the concentration of gallic acid and bromate ions and the potential and deposition time. To create a calibration curve, vanadium was added in the concentration range of 0–2 × 10^−8^ mol L^−1^. Initially, small amounts of vanadium added to the solution showed two peaks; with the increasing VO_2_^(+)^ concentration, a peak at a higher potential became more visible. The formation of two peaks was probably due to the existence of two forms of the complex, one strongly adsorbed by the monolayer directly on the electrode surface and the other one from the weakly adsorbed multilayer. At low concentrations of vanadium, the appearance of two peaks was due to more or less equal amounts of these species in the solution, whereas as the concentration increased, the electrode surface was dominated by the VO_2_^(+)^–GA complex, which caused the multilayer absorption peak to dominate [60].

#### 2.2.6. Catalytic System VO_2_^(+)^–QSA–BrO_3_^−^

In the work [62], it is proposed to increase the detection limit of vanadium by introducing a catalytic effect into adsorptive stripping voltammetry based on the use of quertic–5–sulfonic acid (QSA) as a complexing agent and bromate ions as an oxidizing agent. In this work, it was decided to distinguish the oxidation state of vanadium in which the reduction takes place and in which the signal is obtained. By carrying out measurements on a solution containing QSA as a complexing agent, BrO_3_^−^ as oxidizing ions and the same additions of V(II), V(III) and VO_2_^(+)^ ions, a reduction signal at a potential of −0.76 V was obtained. This made it impossible to distinguish which oxidation state was being reduced, so the measurements were repeated without the presence of bromate ions. The signals obtained were also at the same potential, but the peak heights differed significantly. When bromate ions were added to the solutions containing V(II)–QSA and V(III)–QSA, the color changed to dark yellow, which was the color of the VO_2_^(+)^–QSA solution. Based on the obtained results and observations, the following stages of vanadium determination were established:Oxidation of vanadium(II) and vanadium(III) ions to VO_2_^(+)^, followed by the formation of the VO_2_^(+)^–QSA complex.On the surface of the HMDE working electrode, the VO_2_^(+)^–QSA complex is adsorbed and then reduced to the V(III)–QSA complex.During stripping, the V(III)–QSA complex is reduced to V(II). V(II) is chemically oxidized to VO_2_^(+)^, forming the VO_2_^(+)^–QSA complex, and the reduction is repeated, increasing the signal [62].

### 2.3. Working Electrodes

Commonly used electrodes in adsorptive stripping voltammetry are mercury electrodes. These electrodes operate over a wide range of negative potentials, with excellent repeatability, polarizability and surface smoothness, but the use of mercury electrodes results in the release of mercury into the environment. The increased risks associated with the use, handling and disposal of metallic mercury and its salts required for mercury film production limit the use of mercury in laboratory practice. It is also driving the search for alternative electrodes that use mercury in small amounts or in a safe amalgam form. Most AdSV determinations of trace amounts of vanadium were carried out by using a mercury electrode as the working electrode [79]. Mercury electrodes used in the determination of trace amounts of vanadium included the hanging mercury drop electrode (HMDE) [45,50,52,54,55,56,58,59,62,63,65,66], mercury film electrode (MFE) [46,61], renewable mercury film silver-based electrode (Hg(Ag)FE) [53] and mercury-coated gold microwire electrodes (MWEs) [60]. As you can see, the most frequently chosen electrode for the determination of vanadium was the hanging drop mercury electrode. The design of the hanging drop mercury electrode was proposed by W. Kemula, a professor at the University of Warsaw. The wide application of HMDEs in the analysis of organic compounds and elements results from the large range of potentials in which analyses can be made and the constantly renewable electrode surface, on which there are no products of previous analyses. The use of this electrode in the detection of trace amounts of VO_2_^(+)^ by using the AdSV method was first proposed by Wang et al. In this procedure, the vanadium–cupferron complex accumulated on the hanging mercury electrode, which was followed by a catalytic reduction reaction of the adsorbed complex in the presence of bromate ions. The use of steps such as the adsorption of the complex and the catalytic effect using the HMDE allowed a detection limit that is very impressive to be obtained. Many researchers focused on proposing a VO_2_^(+)^ determination method using an HMDE while obtaining the lowest detection limits and linearity ranges. In the case of using this electrode, unfortunately, a drop of mercury is generated with each measurement, which discourages the use of this vanadium determination procedure [45].

The first attempt to use a mercury film plated on a glassy carbon electrode for the determination of traces of vanadium in environmental samples was proposed by Adeloju and Pablo [61]. The measurements were carried out using the cathodic adsorptive stripping voltammetry (AdCSV) technique. When measurements are carried out without the presence of vanadium or pyrogallol in the samples, there are no visible peaks in the voltammograms, but a peak appears when both substances are present. If the accumulation potential is appropriately chosen, a mercury film is formed in situ on the electrode and adsorption of the complex on the electrode occurs simultaneously. The use of the developed method is encouraged by the lower consumption of mercury and the fact that it is possible to determine low concentrations of vanadium by extending the accumulation time [61].

Another electrode used for the determination of vanadium by cathodic adsorptive stripping voltammetry is the cyclic renewable mercury film silver-based electrode (Hg(Ag)FE) [53]. In order to minimize the use of mercury due to its toxicity, especially in environmental analysis, the use of the Hg(Ag)FE electrode is highly desirable. This electrode uses a very small amount of mercury—only 1 µL of mercury per 1000 cycles of measurement. Comparing the voltammograms obtained with the HMDE and Hg(Ag)FE electrodes, given that the electrode sizes are similar, the signal obtained on Hg(Ag)FE is about 25% higher than that obtained on an HMDE. It should be noted that these experiments used the same measurement techniques and the same ligand, and only different working electrodes were used [53].

Mercury-coated gold microwire electrodes (MWEs) for the determination of VO_2_^(+)^ by the CAdSV method were used for the first time in the study described in the paper [60]. The above catalytic system used VO_2_^(+)^–GA–BrO_3_^−^. The advantage of these electrodes is the overpotential for direct bromate reduction so that catalytic complex detection can be used at potentials higher than the direct bromate reduction potential. MWEs use much less mercury than mercury drop electrodes and offer comparable detection limits. Mercury-coated gold microwire electrodes are also more stable and safer than mercury or foil electrodes because mercury forms an alloy with gold. Therefore, these electrodes are chosen when low detection limits are desired [60].

As a more environmentally friendly method, Wang proposed a procedure for the determination of vanadium using bismuth-coated glassy carbon electrodes, which resulted in eliminating the release of toxic mercury into the environment. Comparing the results obtained for glassy carbon electrodes coated with mercury or bismuth, an increase in the signal using bismuth instead of mercury coating is clearly visible. The measurement procedure was based on the addition of bromate ions to the V–CAA system, thanks to which the peak increased four times. The proposed procedure was successfully used to test a groundwater sample. The obtained results were consistent with the results obtained by atomic absorption spectroscopy [51].

In recent years, a number of mercury-free electrodes have been proposed due to the toxicity of mercury. The following electrodes were used in voltammetric vanadium determination: solid bismuth microelectrode (BiFµE) [47], lead-coated glassy carbon electrode (GCE/PbF) [48], lead film electrode (PbFE) [49], bismuth film electrode (BiFE) [52] and acetylene black paste electrode (ABPE) [57]. The most recently developed method for the determination of vanadium by adsorptive stripping voltammetry using a solid bismuth microelectrode deserves attention here [47]. BiFµE was used to determine both metal ions and organic compounds in 2020 by the research team of I. Gęcy. Using this modern electrode, we are able to eliminate mercury electrodes, thanks to which no toxic mercury is released into the environment. The detection limit for VO_2_^(+)^ using a bismuth microelectrode is sufficient to determine real samples. The glass capillary in which the molten metallic bismuth is hidden is 5 mm thick and 25 µm in diameter, which means that it meets the conditions of a microelectrode. A miniaturized housing made of PEEK is filled with this capillary. Due to the placement of molten bismuth ions in the capillary, it is not necessary to add bismuth ions to the test solution. The use of this electrode simplifies and shortens the analytical procedure and is more friendly to the laboratory environment [47]. Although the vast majority of the working electrodes in AdSV procedures for vanadium determination are mercury based [45,46,50,52,53,54,55,56,58,59,60,61,62,63,64,65,66] and environmentally friendly, electrodes are so far the vast minority [47,48,49,51,57] and represent the future as the basis for green methods. At the same time, as part of the very important Green Analytical Chemistry, they play a major role in making analytical procedures more environmentally benign and safer for humans. Among the criteria dedicated to assess the greenness of voltammetric procedures, the key elements are the working electrodes, such as in the case of vanadium determination BiFµE, BiFE, PbFE, ABPE which, compared to mercury electrodes, significantly reduce the amounts and toxicity of reagents and generated waste and at the same time enable miniaturization and automation. Figure 2 shows the graphical dependence of the number of publications devoted to the determination of V(V) by the AdSV method, depending on the type of material forming the base of the working electrode.

### 2.4. Influence of Interferents on Vanadium Determination

The presence of other ions and organic substances has a huge impact on the signals obtained during the analysis of real samples in stripping voltamperometry. Their presence can affect the signal from the designated ion in a negative way. The factors that can reduce the vanadium signal obtained by adsorptive stripping voltammetry are discussed below.

#### 2.4.1. Influence of Foreign Ions

Both in adsorptive stripping voltammetry and other analytical techniques, the signal is disturbed by the presence of foreign ions in the sample. In the works [45,46,47,48,49,50,51,52,53,55,56,57,58,59,60,61,62,63], the influence on the VO_2_^(+)^ signal caused by the presence of such ions as Ag(I), Co(II), Cd(II), Hg(II), Sr(II), Ca(II), Mg(II), Mn(II), Cu(II), Pb(II), Zn(II), Ni(II), Fe(II), Fe(III), Cr(III), Bi(III), Sb(III), As(III), Ga(III), Zr(IV), Ge(IV), Sn(IV), Te(IV), WO_2_^(2+)^ and UO_2_^(2+)^ was investigated. It can be observed that depending on the type of complexing agent used and the type of working electrode, interferences related to the presence of foreign ions are different. Using the VO_2_^(+)^–cupferron complex accumulated on an MFE, no effect of foreign ions in a 50-fold excess on the obtained vanadium signal was found. This fact proves the high selectivity of this method [46]. However, in the case of the registration of voltammograms by Wang et al., additional peaks were noted in the presence of Ti(IV) and Cu(II) in the solution. In this case, an HMDE was used as the working electrode and cupferron as the complexing agent. The additional titanium and copper peaks did not affect the vanadium peaks [45]. In the vanadium–cupferron system using GCE/PbF as the working electrode, no influence of selected excesses of foreign ions on the vanadium signal was detected. Measurements were carried out in 1000-fold excesses of Ca(II), Mg(II), Bi(III) and Cu(II), and 100-fold excesses of Ni(II), Cd(II) and Fe(III) [48].

Using a hanging drop mercury electrode with 2,3–dihydroxynaphthalene (DHN) as a complexing agent, it was noted that a 200-fold excess of aluminum and tin reduced the signal by 35% and 73%, respectively. A similar effect of MoO_2_^(2+)^–DHN–BrO3– was also visible, which at a lower concentration of molybdenum did not affect the determination of vanadium. However, when the concentration of molybdenum exceeds the concentration of vanadium, interference occurs [63]. The authors of [59] also used an HMDE as a working electrode with 2,3–dihydroxybenzaldehyde as a complexing agent for VO_2_^(+)^ determination. The only observed interference was the lead ion. A 100-fold excess of Pb(II) caused the appearance of a second peak and did not affect the VO_2_^(+)^ peak [59]. Indium turned out to be an interfering substance in the VO_2_^(+)^–catechol system. In the measurement with more than a three-fold excess of In(III), a peak was recorded which superimposed on the VO_2_^(+)^ peak and caused its height to decrease. Fortunately, the concentration of In(III) in seawater was so low that it did not affect the signals obtained when analyzing this type of sample for the presence of VO_2_^(+)^ [55].

It was observed that the VO_2_^(+)^–CAA signal may be reduced due to signal overlap or competition for the surface area on the electrode [51]. Interferences were observed when chloranilic acid was used as a complexing agent. The MoO_2_^(2+)^ and V(VI) peaks appeared on the voltammogram, but this did not affect the VO_2_^(+)^ peak. The MoO_2_^(2+)^ peak was also found to appear in the research conducted by Bobrowski et al. Ions such as Sn(II), Sn(IV) and Cd(II) in large excess can interfere with the vanadium signal due to the low ratio of the diffusion current to the catalytic current VO_2_^(+)^. There is a risk of signal interference in the presence of antimony because the Sb(III)–CAA complex is adsorbed on the electrode surface and the Sb(III) peak appears at the same potential as the VO_2_^(+)^ peak. The determination of vanadium via the AdSV method interferes with a ten-fold excess of antimony ions, while in the case of the DPP method, it interferes with a four-fold excess. To eliminate or minimize the interference from interfering agents, the authors suggested recording polarographic or voltammetric curves in the presence and absence of bromate ions [52].

A voltammogram in the presence of antimony ions was also recorded. Chloranilic acid forms a complex with Sb(III), which causes competition on the surface of the working electrode with the CAA–chloranilic acid complex. Already a 10-fold excess of Sb(III) will cause a decrease in the V signal [52]. During the determination of vanadium using Hg(Ag)FE as the working electrode, many signal interferences occurred. The ions that did not interfere with the assay were Fe(III), Zn(II) and Mn(II). Pb(I), Sn(II), Cd(II), UO_2_^(2+)^, MoO_2_^(2+)^, WO_2_^(2+)^, Se(IV), Cu(II) and Bi(III) were the ions that caused the signal to drop. The addition of a 20-fold excess of Pb(II) and Sn(II) ions produced the most visible effect as the vanadium signal decreased by up to 70%. The vanadium signal decreased in height by 90% as a result of a 100-fold excess of MoO_2_^(2+)^. As a result of such a large excess, WO_2_^(2+)^ had the smallest effect [53]. When registering the voltammogram for vanadium in [60], lead and cadmium turned out to be interfering substances. To minimize their impact, EDTA should just be added, which will form a complex with them [61].

#### 2.4.2. Influence of the Organic Matrix of Samples

In the case of vanadium determination, organic substances may influence the AdSV signal. Organic matter can accumulate on the surface of the working electrode, which blocks its surface and prevents the accumulation of the vanadium complex. Vanadium interference effects are shown below. The following substances were used in the research: humic acids (HA), fulvic acids (FA), cetyltrimethylammonium bromide (CTAB—cationic surfactant), sodium Dodecyl Sulfate (SDS—anionic surfactant) and Triton X–100 (nonionic surfactant).

Interferences from surfactants occurred using the following measurement setups. The tested measurement systems include VO_2_^(+)^–cupferron [47,48,49,50], VO_2_^(+)^–CAA [51,52,53], VO_2_^(+)^–catechol [55], VO_2_^(+)^–CCR [56], VO_2_^(+)^–alizarin violet [57], VO_2_^(+)^–pyrocatechol violet [58], VO_2_^(+)^–2,3–DHBA–BrO_3_^−^ [59], VO_2_^(+)^–GA–BrO_3_^−^ [60], VO_2_^(+)^–pyrogallol [61], VO_2_^(+)^–QSA–BrO_3_^−^ [62] and VO_2_^(+)^–2,3–DHN–BrO_3_^−^ [63]. When humic acids were added to the samples in the amounts of 0.025 mg L^−1^ and 0.5 mg L^−1^, 25% attenuation of the signal was visible for the first addition, while in the case of the second one, the vanadium signal disappeared completely [53]. Triton X–100 decreased the height of the VO_2_^(+)^ peak by 40% and 90% at the positive time point, in the first case by 0.05 mg L^−1^ and 0.2 mg L^−1^, respectively [52]. A similar effect of Triton X–100 was observed in the research led by Professor Tyszczuk. The only difference was a higher content of nonionic surfactant added to the solution, as much as 0.5 mg L^−1^. It also reduced the value of the vanadium signal by 90% [48]. Surfactants such as Triton X–100 and SDS were also shown to have a negative influence on the VO_2_^(+)^ signal. To eliminate the influence of interfering substances, it was necessary to add nitric acid to the solution to be determined and to heat the sample dry [54].

In order to limit the influence of surfactants, proper sample preparations should be applied before taking the measurements. In the case of vanadium determination, the below-mentioned solutions were proposed to remove or reduce interference. Before the measurement, the sample should be exposed to the ultra-trace radiation or mineralization of the sample. In the work [49], a new effective method of eliminating or reducing interference caused by the presence of organic substances was also introduced. This procedure involved mixing the sample with Amberlite XAD–7 resin. In this case, 0.5 g of resin was mixed with the test solution. During the mixing, the organic substances were adsorbed on the resin surface, so they did not interfere with the measurements of the vanadium signal.

## 3. Simultaneous Determination of Vanadium Ions with Other Ions

The advantage of AdSV is the possibility to determine two or more ions simultaneously. By using stripping voltammetry techniques, we are able to determine multiple ions in a single measurement. Sylvia Sander decided to determine four elements simultaneously using chloranilic acid as a ligand. The method used for the determination of vanadium simultaneously with molybdenum, antimony and uranium was adsorptive stripping voltammetry (AdSV). The research was carried out in a three-electrode voltammetric cell with the use of a hanging mercury electrode as a working electrode. No catalytic effect was used in the assay procedure. After selecting the appropriate concentration of chloranilic acid, solution pH, differential pulse amplitude and accumulation potential, real samples were tested for the presence of MoO_2_^(2+)^, UO_2_^(2+)^, VO_2_^(+)^ and Sb(III) ions [54].

In the work [56] using CCR as a complexing agent, the simultaneous determination of molybdenum and vanadium was proposed. Before optimizing the parameters, they checked whether signals appeared on the voltammogram in the presence of only CCR. No visible peak was found for CCR, but when VO_2_^(+)^ at 3 ng mL^−1^ was added, a peak appeared at −0.16 V and a MoO_2_^(2+)^ peak at 7 ng mL^−1^ at −0.47 V. The effect of foreign ions on the signal of vanadium and molybdenum was also investigated, and no interference from foreign ions was found, except for thiosulfide ions [56].

The use of a mixture of two ligands in cathodic adsorptive stripping voltammetry was proposed by Cobelo–Garcia et al. The method for the simultaneous determination of traces of the metals copper, nickel and vanadium was optimized [65]. Abbasi et al. proposed adsorptive voltammetry using a hanging drop mercury electrode for the simultaneous determination of traces of lead and vanadium. They focused on optimizing the procedure to determine the concentration of the ligand cupferron. The parameters of the process, such as the potential, accumulation time and scanning speed, but also the appropriate supporting electrolyte and its pH, were selected. They also showed that the developed method is selective with respect to the coexistence of impurities as they do not affect the signal of vanadium and lead. The results obtained are consistent with the results obtained using the reference method. This method allows for the simultaneous determination of vanadium and lead in a fast, inexpensive, sensitive and selective manner [50].

Cathodic adsorptive stripping voltammetry was used for the simultaneous determination of cobalt, copper, nickel, iron and vanadium. To determine these five elements in pore water samples, Santos–Echeandía used a mixture of DMG and catechol as a complexing agent. The selection of such a complexing agent mixture resulted in two additional peaks being obtained in the determination of copper, nickel and vanadium. After a thorough analysis, the author concluded that these elements were iron and cobalt, as we can see according to literature sources. The selection of appropriate ligand concentrations was complicated by the slow formation of Co–DMG and Ni–DMG complexes. The simultaneous determination of these ions required the presence of 0.6 mmol L^−1^ DMG and 0.9 mmol L^−1^ catechol in the solution to obtain the highest sensitivity of the method [66].

An interesting application is the use of cupferron, oxalic acid and 1,3–diphenylguanidine (COD mixture) as the working solution. A method for the simultaneous determination of Zr(IV) and VO_2_^(+)^ by voltammetry using a hanging drop mercury electrode and a COD mixture was developed. It is necessary to use sequential rather than simultaneous element determination because the concentration of vanadium ions in real samples is one or two orders of magnitude higher than that of zirconium ions. Initially, the signal is tested for increasing concentrations of Zr(IV), and then measurements are made to change concentrations of VO_2_^(+)^. The determination of vanadium and zirconium can be affected by the formation of Ti(IV)–COD and Hf(IV)–COD complexes, but these interferences do not weaken the signal of the determined ions in real samples. The results obtained show that it is possible to use this method on natural water samples, but it is necessary to expose them to UV radiation [64].

## 4. Application

On the basis of Table 3 and Table 4, which summarize the practical application of the reviewed procedures in food and environmental samples, respectively, it is possible to assess their suitability for the determination of vanadium and other elements by adsorptive stripping voltammetry. We can use adsorptive stripping voltammetry procedures to determine ions in a wide range of samples. Vanadium ions have been tested in samples of food, tap water, rainwater, groundwater, environmental water, river water and in certified reference materials.

The paper [65] presents the possibility of using the developed AdCSV method for the simultaneous determination of copper, nickel and vanadium in samples taken from the coast of Spain. The AdSV method using an HMDE as a working electrode and a mixture of DMG ligands and catechol as a complexing agent has been proven to be a suitable method for controlling vanadium concentrations in the environment. Water samples from the Vigo Rio estuary were used to test the suitability of the method. The tanker Prestige, which was carrying large amounts of fuel oil, sank in Spanish waters. In order to check whether oil leaking into the water had an effect on the increase in the Cu concentration, the tests were carried out using the AdCSV method. The tests showed that the concentrations of the determined elements were comparable to the concentrations tested in the uncontaminated area. This proves the usefulness of the proposed procedure for environmental analyses [65].

It is possible to test food and water samples for the presence of vanadium and lead ions using the AdSV method. The legitimacy of using the proposed method for the analysis of real samples can be assessed by the results obtained. In the short duration of the complex accumulation stage on the surface of the working electrode, which was an HMDE, the same results were obtained as in the method of atomic absorption spectroscopy. Both lead and vanadium ions were detected in the tested food, with the highest content of vanadium found in tomatoes and the lowest one in flour [50].

As natural samples, NASS sea water, sea water from the North Fijian sea, local drinking water and a sewage sample from a suspended uranium slag heap were selected. For the determination of molybdenum, vanadium, antimony and uranium ions in these samples, adsorptive stripping voltammetry was used using chloranilic acid as a complexing agent. It was proven that all four elements can be determined in drinking or spring waters without prior sample preparation before measurements. Fresh surface water samples, organically polluted water samples or wastewater samples from suspended uranium slag heaps must be properly prepared before measurement. The developed procedure for the determination of MoO_2_^(2+)^, UO_2_^(2+)^, VO_2_^(+)^ and Sb(III) ions by the AdSV method is one order of magnitude smaller for seawater than freshwater samples. The reason for this is the greater presence of chlorides, which interact with mercury and cause a change in the structure of the HMDE electrode [54].

The application of the developed method for the simultaneous determination of vanadium and molybdenum using the AdCSV method and an HMDE as the working electrode was described in the paper [56]. The authors performed measurements on real samples, including tap water, river water and well water, as well as food products such as tomatoes, cucumbers, carrots and tea. They compared the results obtained with the ICP method to demonstrate the usefulness of the AdCSV method. The signals obtained by the AdCSV method make it possible to determine its usefulness and the possibility of using this method for the determination of vanadium and molybdenum ions [56].

**Table 3 materials-16-03646-t003:** Simultaneous determination of vanadium and other metal ions in food by AdSV and reference AAS and ICP–OES methods.

Complexing Agent	Sample	Found VO_2_^(+)^ by AdSV Method		Found VO_2_^(+)^ by AAS Method		Ref.
		Pb(II) (ng g^−1^)	VO_2_^(+)^ (ng g^−1^)	Pb(II) (ng g^−1^)	VO_2_^(+)^ (ng g^−1^)	
Cupferron	Potato	231.37 ± 0.64	440.38 ± 1.21	233.02 ± 1.63	439.34 ± 1.37	[50]
	Rice	2.40 ± 0.13	115.98 ± 1.15	2.96 ± 0.28	116.57 ± 1.01	
	Flour	10.81 ± 0.36	98.06 ± 0.56	10.17 ± 0.49	96.97 ± 0.79	
Chromoxane cyanine R	**Sample**	**Found VO_2_^(+)^ by** **AdSV Method**		**Found VO_2_^(+)^ by** **ICP–OES Method**		
		**MoO_2_^(2+)^Mo(VI)** **(ng mL^−1^)**	**VO_2_^(+)^** **(ng mL^−1^)**	**MoO_2_^(2+)^** **(ng mL^−1^)**	**VO_2_^(+)^** **(ng mL^−1^)**	
	Tomato	2.54 ± 0.08	1.06 ± 0.04	2.50 ± 0.06	1.05 ± 0.05	[56]
	Carrot	1.92 ± 0.04	0.76 ± 0.05	1.95 ± 0.04	0.76 ± 0.05	
	Tea	3.41 ± 0.13	1.53 ± 0.06	3.40 ± 0.09	1.55 ± 0.06	
	Cucumber	1.86 ± 0.06	0.96 ± 0.05	1.90 ± 0.06	0.98 ± 0.04	

**Table 4 materials-16-03646-t004:** Determination of vanadium and other metal ions in environmental samples by the AdSV method.

Complexing Agent	Sample	Added VO_2_^(+)^ (nmol L^−1^)	Found VO_2_^(+)^ (nmol L^−1^)				Ref.
Cupferron	Ciemiega river water	5.0 10.0	5.2 9.9				[47]
	Tap water	5.0 10.0	4.9 9.6				
	Rainwater	5.0 10.0	5.3 10.4				
Cupferron	Vistula River	- 5.0 10.0	4.7 10.2 14.5				[49]
Chloranilic acid	Vistula river	- 2.5 5.0 10.0	6.47 ± 0.31 9.06 ± 0.37 11.1 ± 0.4 –				[53]
	Rudawa river	- 2.5 5.0 10.0	5.27 ± 0.11 7.61 ± 0.16 10.5 ± 0.2 –				
	Wilga river	- 2.5 5.0 10.0	3.17 ± 0.16 5.44 ± 0.22 8.09 ± 0.23 –				
	Snow	- 2.5 5.0 10.0	14.3 ± 0.4 – 18.7 ± 0.5 25.0 ± 0.5				
Alizarin violet	Tap water	-	9.87				[57]
	River water	-	22.5				
	Mineral water	-	9.10				
Cupferron	Ilam city water	**Added VO_2_^(+)^ (ng mL^−1^)**	**Found VO_2_^(+)^** **(ng mL^−1^)**				[50]
		- 10.0 30.0	0.48 ± 0.08 11.01 ± 0.55 30.99 ± 0.24				
	Zayandehrood river water	- 10.0 30.0	0.91 ± 0.18 10.54 ± 0.30 30.75 ± 0.24				
	Mineral water	- 10.0 30.0	0.15 ± 0.09 10.41 ± 0.21 30.85 ± 0.20				
	Ilam University laboratory water	- 10.0 30.0	– 10.08 ± 0.12 30.38 ± 0.19				
Chromoxane cyanine R	River water	- 2.0 5.0	5.67 ± 0.17 7.57 ± 0.20 10.51 ± 0.32				[56]
	Tap water	- 2.0 5.0	– 1.96 ± 0.10 4.88 ± 0.15				
	Well water	- 2.0 5.0	– 1.94 ± 0.10 4.93 ± 0.12				
**Complexing Agent**	**Sample**	**Certified Reference**	**Found VO_2_^(+)^**				**Ref.**
Cupferron	SPS–WW1 Wastewater	100.0 ± 0.5 (ng mL^−1^)	96.2 ± 1.7 (ng mL^−1^)				[48]
Cupferron	CRM (estuarine water)	0.51 ± 0.61 (nmol L^−1^)	0.496 ± 0.51 (nmol L^−1^)				[49]
Pyrogallol	Urban particulate matter 1648	140 ± 3 (µg L^−1^)	137.3 ± 7.7 (µg L^−1^)				[61]
	Peach leaves 1547	0.37 ± 0.03 (µg L^−1^)	0.34 ± 0.04 (µg L^−1^)				
	Apple leaves 1515	0.26 ± 0.03 (µg L^−1^)	0.25 ± 0.05 (µg L^−1^)				
	Bovine liver 1577b	0.123 (µg L^−1^)	0.117 ± 0.020 (µg L^−1^)				
**Complexing Agent**	**Sample**	**Found**					**Ref.**
		**Ur(VI)** **(µg L^−1^)**	**Sb(III)** **(µg L^−1^)**	**MoO_2_^(2+)^** **(µg L^−1^)**	**VO_2_^(+)^** **(µg L^−1^)**		
Chloranilic acid	NASS sea water standard	2.86 ± 0.43	–	8.28 ± 0.25	–		[54]
	Sewage of uranium slag heap	14.03 ± 1.98	–	2.650 ± 0.40	6.968 ± 2.20		
DMG+Catechol		**Cu(II)** **(nmol L^−1^)**		**Ni(II)** **(nmol L^−1^)**	**VO_2_^(+)^** **(nmol L^−1^)**		[65]
	seawater reference material CASS–4	9.7 ± 0.9		9.7 ± 0.9	9.7 ± 0.9		
DMG+Catechol		**Co(II)** **(nmol L^−1^)**	**Cu(II)** **(nmol L^−1^)**	**Fe(III)** **(nmol L^−1^)**	**Ni(II)** **(nmol L^−1^)**	**VO_2_^(+)^** **(nmol L^−1^)**	[66]
	seawater reference material CASS–4	0.56 ± 0.08	8.5 ± 0.8	14 ± 3	5.9 ± 0.5	19 ± 2	

## 5. Conclusions

In conclusion, this has been a review of the use of the adsorptive stripping voltammetry method for the determination of vanadium. The amount of researchers that have used adsorptive stripping voltammetry to determine vanadium can be seen in the number of papers devoted to this issue presented in Table 1. Based on the collected articles on the subject, the factors influencing the obtained signal have been discussed. The key element in the AdSV procedures is the selection of a complexing agent; an analysis of the collected works was carried out in this respect, and it was found that by far the most commonly used complexing agents were cupferron and chloranilic acid. It was cupferron that was used in the procedure that gave the lowest detection limit, i.e., 2.8 × 10^−12^ mol L^−1^ [48]. In order to lower the limit of detection in many procedures, various catalytic systems based on the addition of oxidizing agents to the solution were introduced for measurements, among which bromate ions were proven to be the best choice. In the case of vanadium determination, the following catalytic systems were used: VO_2_^(+)^–cupferron–BrO_3_^−^, VO_2_^(+)^–chloranilic acid–BrO_3_^−^, VO_2_^(+)^–catechol–BrO_3_^−^, VO_2_^(+)^–2,3–DHBA–BrO_3_^−^, VO_2_^(+)^–GA–BrO_3_^−^, VO_2_^(+)^–QSA–BrO_3_^−^ and VO_2_^(+)^–DHN–BrO_3_^−^. Comparing the detection limits obtained using the same working electrode and complexing agent, it can be seen that the addition of an oxidizing agent reduced the limit by as much as two orders of magnitude [45,50]. Another key element in voltammetric procedures is the type of working electrode used, on which the reaction that is the basis of the measurement takes place. Scientists are increasingly trying to introduce new working electrodes that can be used for the determination of vanadium, as can be seen from the summary in Table 1. However, mercury-based electrodes are by far the most commonly used electrodes. It should be noted, however, that in recent years, several voltammetric procedures for the determination of VO_2_^(+)^ have emerged using more environmentally friendly mercury-free electrodes based mainly on bismuth and lead, which constitute a promising starting point for the search for new electrode materials that fit into Green Analytical Chemistry. It is worth emphasizing that in almost all AdSV procedures dedicated to the determination of vanadium, the influence of many foreign ions as potential interfering elements on the analytical signal of vanadium was examined. Their influence obviously depended on the working electrode and the complexing agent used in a given procedure. The interfering ions included titanium, copper, tin, lead, indium, molybdenum and antimony ions, and the interferences associated with them most often resulted from the appearance of additional peaks on the voltammogram. Unfortunately, fewer studies have examined the effect of surface-active substances on the vanadium signal, and these studies were most often conducted by examining the change in the vanadium signal in the presence of Triton X-100 as a representative of surfactants, which caused a significant decrease in the vanadium signal. In an even smaller number of works, specifically merely in three, the effect of humic substances was examined, which also, as proved, caused a decrease in the vanadium peak. At the end of the review of the AdSV procedures for the determination of vanadium in Table 3 and Table 4 presented in this paper, their practical application was collected by analyzing real samples, such as food, soil and most often environmental water. To sum up, it can be said that this review of AdSV procedures for the determination of vanadium is a good compendium that summarizes their analytical characteristics and applicability.

## Figures and Tables

**Figure 1 materials-16-03646-f001:**
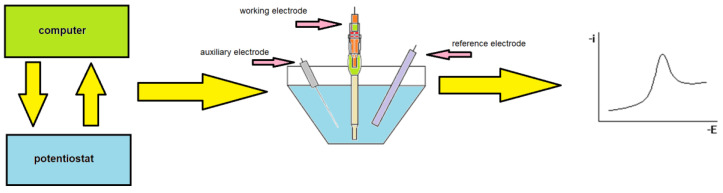
A schematic diagram illustrating the course of voltammetric measurements.

**Figure 2 materials-16-03646-f002:**
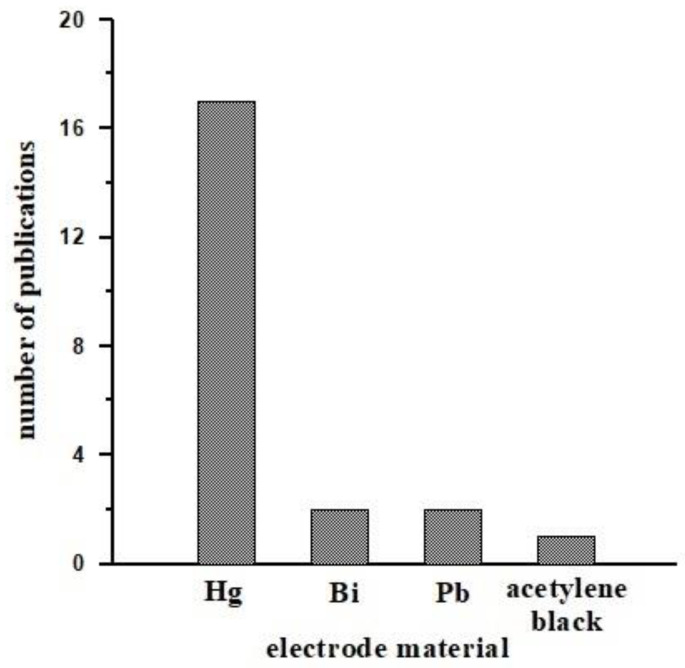
The number of publications devoted to the determination of V(V) by the AdSV method, depending on the electrode material.

**Table 2 materials-16-03646-t002:** Chemical structures of the most commonly used complexing agents while using the AdSV method of determining vanadium.

Complexing Agent	Chemical Structure	Ref.
Cupferron	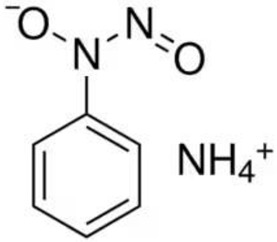	[72]
Chloranilic acid	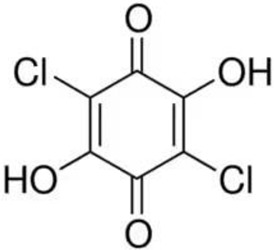	[73]
Catechol	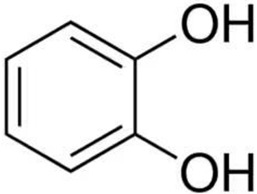	[74]
Gallic acid	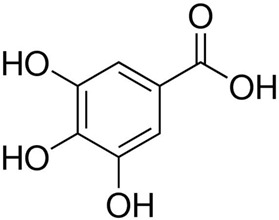	[75]
2,3–dihydroxybenzaldehyde	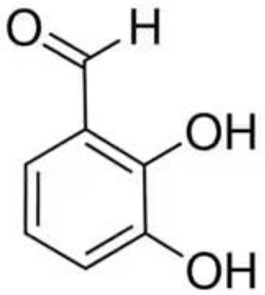	[76]
2,3–dihydroxynapthlhalene	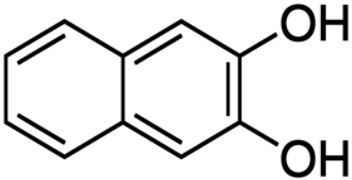	[77]

## Data Availability

Not applicable.

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
