# Peer review of "Adsorptive Stripping Voltammetry for Determination of Vanadium: A Review"

_materials, 2023, doi:10.3390/ma16103646_

Round 1
Reviewer 1 Report
This paper provides an overview of methods of adsorptive stripping voltammetry that can be used to determine trace amounts of V(V) in various types of samples. However, this paper seem more like a simple report than a helpful review of this field, and it is poorly scholarly. Besides, there are only 41 references cited, I do not believe that the authors have done a good job of summarizing the progress of research in the field. So I do not recommend this paper for publication in Materials.
Author Response
Thank you for your letter and the reviewers’ reports of my paper entitled: “A review on adsorptive stripping voltammetric procedures for the determination of vanadium”. Now I am sending you a revised version of the manuscript, corrected according to the reviewer’s suggestions. The changes made in the revised manuscript are highlighted.
The revisions which we made are as follows:
Reviewer 1:
Remark: This paper provides an overview of methods of adsorptive stripping voltammetry that can be used to determine trace amounts of V(V) in various types of samples. However, this paper seem more like a simple report than a helpful review of this field, and it is poorly scholarly. Besides, there are only 41 references cited, I do not believe that the authors have done a good job of summarizing the progress of research in the field. So I do not recommend this paper for publication in Materials.
Answer:
The manuscript has been significantly expanded, additional information related to vanadium analysis has been added, new figures have been added and the literature has been supplemented to 79 references.
Yours sincerely,
Prof. Malgorzata Grabarczyk
Reviewer 2 Report
The manuscript is a review article entitled “A review on adsorptive stripping voltammetric procedures for the determination of vanadium” Electrochemical sensors have attracted a great deal of attention for their applications. Before accepting it for publication, the following points need to be addressed:
1) Since both authors have the same affiliation, please combine them into one affiliation and remove the numbers from their affiliations.
2) Besides electrochemistry, other available techniques must be compared in detail to determine their advantages and disadvantages.
3) You should include Table 1 after the first citation in the text.
4) The introduction should include a general description of the adsorptive stripping voltammetry technique from the electrochemistry viewpoint (I would appreciate it if you could provide some figures to understand this technique better).
5) Figures from published articles must be included in the review, and both the authors and the publishers must give their permission.
6) It is important to add figures to make the review more understandable for readers. You should include figures in every section of your review.
As of now, I need to receive a response from the authors before making a decision.
Author Response
Thank you for your letter and the reviewers’ reports of my paper entitled: “A review on adsorptive stripping voltammetric procedures for the determination of vanadium”. Now I am sending you a revised version of the manuscript, corrected according to the reviewer’s suggestions. The changes made in the revised manuscript are highlighted.
The revisions which we made are as follows:
Reviewer 2:
Remark: 1) Since both authors have the same affiliation, please combine them into one affiliation and remove the numbers from their affiliations.
Answer: The affiliations have been combined into one.
Remark: 2) Besides electrochemistry, other available techniques must be compared in detail to determine their advantages and disadvantages.
Answer: Information on other available techniques used to determine vanadium has been added in Introduction.
Remark: 3) You should include Table 1 after the first citation in the text.
Answer: Table 1 has been moved to Section 1.
Remark: 4) The introduction should include a general description of the adsorptive stripping voltammetry technique from the electrochemistry viewpoint (I would appreciate it if you could provide some figures to understand this technique better).
Answer: Text describing adsorption stripping voltammetry techniques has been added to Introduction and Figure 1 showing the schematic process of voltammetric measurement has been added to this Section.
Remark: 5) Figures from published articles must be included in the review, and both the authors and the publishers must give their permission.
Answer:
Unfortunately, due to the short time available, we were unable to obtain permission from authors and publishers to include figures from published articles.
Remark: 6) It is important to add figures to make the review more understandable for readers. You should include figures in every section of your review.
Answer: Figures have been included in the work. The added figures can be found in the following places in the work: in Section 1 a schematic diagram illustrating the voltammetric measurements and in Section 2.3 a graphical representation of the dependence of the number of publications devoted to the determination of V(V) by the AdSV method on the type of material underlying the working electrode. Table 2, which shows the chemical structures of the most commonly used complexing agents in the determination of vanadium, is included in the paper.
Yours sincerely,
Prof. Malgorzata Grabarczyk
Reviewer 3 Report
The paper describes the detection of vanadium. Several errors (tipos) that have to be corrected. The manuscript seems to be suitable for publishing if revised (minor revision) according to the comments presented below.
1- Why the authors selected vanadium?
2- In the literature review, the author should be mention the following work:
· Voltammetric Determination of Hg2+, Zn2+, and Pb2+ Ions Using a PEDOT/NTA-Modified Electrode.. https://pubs.acs.org/doi/full/10.1021/acsomega.2c02682, https://doi.org/10.1021/acsomega.2c02682
· Use of a Schiff base-modified conducting polymer electrode for electrochemical assay of Cd (II) and Pb (II) ions by square wave voltammetry.. https://link.springer.com/article/10.1007/s11696-021-01882-7 , https://doi.org/10.1007/s11696-021-01882-7
Author Response
Thank you for your letter and the reviewers’ reports of my paper entitled: “A review on adsorptive stripping voltammetric procedures for the determination of vanadium”. Now I am sending you a revised version of the manuscript, corrected according to the reviewer’s suggestions. The changes made in the revised manuscript are highlighted.
The revisions which we made are as follows:
Reviewer 3:
Remark: 1) Why the authors selected vanadium?
Answer: As written in Introduction vanadium is an essential element for normal cell growth in small amounts, but can be toxic when present in higher concentrations. Additionally both species V(IV) and V(V) have different nutritional and toxic characteristics. Therefore, the accurate determination of vanadium species in different oxidation states is very important. At the same time, due to the narrow threshold value between the necessary and toxic concentrations for living organisms, the determination of traces of vanadium in various samples is very important and meaningful from the point of view of environmental science and life science.
There are many review papers in the literature on the determination of various elements using different methods. However, there has not yet been a review in the literature of procedures devoted to the determination of V(V) by the AdSV method, which is often used for this purpose. Therefore, the aim of this work was to provide such a review.
Remark: 2) In the literature review, the author should be mention the following work:
- Voltammetric Determination of Hg2+, Zn2+, and Pb2+Ions Using a PEDOT/NTA-Modified Electrode.https://pubs.acs.org/doi/full/10.1021/acsomega.2c02682, https://doi.org/10.1021/acsomega.2c02682
- Use of a Schiff base-modified conducting polymer electrode for electrochemical assay of Cd (II) and Pb (II) ions by square wave voltammetry.
https://link.springer.com/article/10.1007/s11696-021-01882-7 , https://doi.org/10.1007/s11696-021-01882-7
Answer: The works mentioned were added in the manuscript as [67] and [68].
Yours sincerely,
Prof. Malgorzata Grabarczyk
Reviewer 4 Report
Referee's Comments on:
“A review on adsorptive stripping voltammetric procedures for the determination of vanadium”
(by E. Wlazłowska & M. Grabarczyk; submitted to Materials (MDPI) and registered
as # MATER-23-19500)
The review article by E. Wlazłowska & M. Grabarczyk summarises recent advances in the development and use of various electroanalytical methods for the determination of vanadium using adsorptive stripping voltammetry (in further text abbreviated as AdSV). It concerns an interesting and actual topic and even in a view of this brief characterisation seems to be worthy of considering for potential publication in Materials (MDPI).
Regarding the manuscript as such, it is structured and written quite well, but there are still numerous parts that require certain improvement or even such places that miss inevitable information. The latter is the case of the electrochemistry of vanadium that is reviewed insufficiently (see below and my addressed comments, especially point 3) By summing up, the paper by E. Wlazłowska & M. Grabarczyk can be recommended for publication after major revision, reflecting the following points:
Addressed Comments:
1) Title // Formal note + Recommendation … In accordance with usual convention, I would recommend its slight reformulation into: "Adsorptive stripping voltammetry for determination of vanadium. A review"; it means the word "review" should not be the part of the title itself. And, also, the term "procedures" seems to be superfluous – if somebody uses AdSV, it is obvious that he/she will need to follow the respective procedure…
2) Abstract // Critical comment + Correction … (A) Almost one third of the abstract looks like a summary for review about the vanadium itself. In other words, the text at first six and half lines (up to "… human body.") should be removed and the new abstract started from afterwards. (B) To my knowledge, vanadium is not heavy metal, but it is being sorted among the so-called strategic metals (together with Fe, Co, Ni, Mo, and W). Please, check it and eventually correct properly.
3) Introduction // Formal note + Principal critical comment… (A) The text starting to introduce electroanalytical techniques and methods for vanadium should begin as a new paragraph (from line 60). (B) In this separate paragraph, some text should be definitely devoted to the reason(s) on why vanadium is being determined by using AdSV and not by means of direct redox transformations. Especially in the case of vanadium, this explanation must be given because this metal is fairly electroactive and capable of occurring in several (cascade) oxidation states – from pentavalent V(V) via V(IV) and V(III), down to bivalent ions, V(II). And if some species are electroactive, then ¾ in electroanalytical methods ¾, the corresponding oxidations / reductions are utilised primarily. However, if there are some problems with such redox reactions; then, indirect procedures employing complexation, ion-pairing, catalysis, etc. are coming to the fore being applicable in the AdSV mode.
4) Chapters 2 and 3: AdSV of Vanadium and the WE; pages 2-10, The individual notes // (A) Section 2.1 … The title is not convenient and should be changed – e.g., into "Methods / Procedures Based on Complexation Reactions"
(B) Sections 2.1 and 2.2, as well as further in the text … The authors repeatedly quote the single ions with very high valences, namely V(V), V(IV), W(VI), Mo(VI), and U(VI). This is incorrect because such ions do not exist (!) The authors should either write about central atoms with such a valence or quote the original anions or, eventually, cationic species binding oxygen, e.g., vanadyl VO2(+) and uranyl UO2(2+). The same applies to some reaction schemes like that in section 2.2 (see also later and my other comment).
(C) Section 2.1 … Some reaction scheme showing the pathway of complexation could be given, similarly like that in the next section 2.2.
(D) Section 2.2 … Please, correct the reaction scheme and the respective formulas in the text according to my comment (4B).
(E) Section 2.3 … It is a fact that a majority of methods for AdSV of vanadium had employed the mercury-based working electrodes, i.e., HMDE and MFE. However, in the optics of the current trends almost refusing mercury as such, this should be commented with respect to possible alternative electrode materials, when pointing out the already existing or methods / procedures employing non-mercury electrodes. In other words, to emphasize more the "green" methods like that commented on page 7, within lines 332-340.
5) Applications and Conclusions; pages 11-12 // Comment … The cardinal problem with the use of mercury-based electrodes I have highlighted in the previous point (4E) should be commented, in a general way, also here, discussing in more detail possible approaches or even prospects on how to replace mercury working electrodes by less-controversial alternatives.
6) Table(s) 1-3; pages 13-20 // Some minor corrections (throughout the data in the tables) …
(A) I recommend the authors to exchange the order of the first and second columns;
(B) Again, replace the highly valent single ions (like V(V) etc.) with the corresponding anions or cationic complexes [like VO3(-) or VO2(+)];
(C) Should be: "chloranilic acid", "…-aldehyde", "ICP-MS" (or "ICP-OES" and not only "ICP", which is the way of ionisation only and not a detection technique / method).
Author Response
Thank you for your letter and the reviewers’ reports of my paper entitled: “A review on adsorptive stripping voltammetric procedures for the determination of vanadium”. Now I am sending you a revised version of the manuscript, corrected according to the reviewer’s suggestions. The changes made in the revised manuscript are highlighted.
The revisions which we made are as follows:
Reviewer 4:
Remark: 1) Title // Formal note + Recommendation … In accordance with usual convention, I would recommend its slight reformulation into: "Adsorptive stripping voltammetry for determination of vanadium. A review"; it means the word "review" should not be the part of the title itself. And, also, the term "procedures" seems to be superfluous – if somebody uses AdSV, it is obvious that he/she will need to follow the respective procedure…
Answer: The title has been changed as indicated.
Remark: 2) Abstract // Critical comment + Correction … (A) Almost one third of the abstract looks like a summary for review about the vanadium itself. In other words, the text at first six and half lines (up to "… human body.") should be removed and the new abstract started from afterwards. (B) To my knowledge, vanadium is not heavy metal, but it is being sorted among the so-called strategic metals (together with Fe, Co, Ni, Mo, and W). Please, check it and eventually correct properly.
Answer:
(A) The unnecessary text in the Abstract has been removed.
(B) The term 'heavy metal' has been removed.
Remark: 3) Introduction // Formal note + Principal critical comment… (A) The text starting to introduce electroanalytical techniques and methods for vanadium should begin as a new paragraph (from line 60). (B) In this separate paragraph, some text should be definitely devoted to the reason(s) on why vanadium is being determined by using AdSV and not by means of direct redox transformations. Especially in the case of vanadium, this explanation must be given because this metal is fairly electroactive and capable of occurring in several (cascade) oxidation states – from pentavalent V(V) via V(IV) and V(III), down to bivalent ions, V(II). And if some species are electroactive, then ¾ in electroanalytical methods ¾, the corresponding oxidations / reductions are utilised primarily. However, if there are some problems with such redox reactions; then, indirect procedures employing complexation, ion-pairing, catalysis, etc. are coming to the fore being applicable in the AdSV mode.
Answer:
(A) The text starting to introduce electroanalytical techniques and methods for vanadium begins as a new paragraph.
(B) In Introduction, it is explained why vanadium is determined by the AdSV method. Information on the ASV determination of vanadium based on oxidation/reduction reactions has been added.
Remark: 4) Chapters 2 and 3: AdSV of Vanadium and the WE; pages 2-10, The individual notes // (A) Section 2.1 … The title is not convenient and should be changed – e.g., into "Methods / Procedures Based on Complexation Reactions"
(B) Sections 2.1 and 2.2, as well as further in the text … The authors repeatedly quote the single ions with very high valences, namely V(V), V(IV), W(VI), Mo(VI), and U(VI). This is incorrect because such ions do not exist (!) The authors should either write about central atoms with such a valence or quote the original anions or, eventually, cationic species binding oxygen, e.g., vanadyl VO2(+) and uranyl UO2(2+). The same applies to some reaction schemes like that in section 2.2 (see also later and my other comment).
(C) Section 2.1 … Some reaction scheme showing the pathway of complexation could be given, similarly like that in the next section 2.2.
(D) Section 2.2 … Please, correct the reaction scheme and the respective formulas in the text according to my comment (4B).
(E) Section 2.3 … It is a fact that a majority of methods for AdSV of vanadium had employed the mercury-based working electrodes, i.e., HMDE and MFE. However, in the optics of the current trends almost refusing mercury as such, this should be commented with respect to possible alternative electrode materials, when pointing out the already existing or methods / procedures employing non-mercury electrodes. In other words, to emphasize more the "green" methods like that commented on page 7, within lines 332-340.
Answer:
(A) Section 2.1. The title was changed.
(B) In line with the suggested comment regarding the notation of the following ions V(V), V(IV), W(VI), Mo(VI), and U(VI) in the work in Chapters 2.1 and 2.2, their notation has been changed.
(C) Unfortunately, the cited works (Section 2.1.) do not present diagrams showing the pathway of complexation.
(D) In section 2.2, the reaction scheme and the corresponding formulas in the text have been revised according to the reviewer's advice.
(E) In section 2.3, additional information has been added to support the fact that mercury electrodes are being phased out in exchange for the introduction of new non-mercury working electrodes for vanadium determination procedures. The information considering “green” procedures of vanadium determination were added.
Remark: 5) Applications and Conclusions; pages 11-12 // Comment … The cardinal problem with the use of mercury-based electrodes I have highlighted in the previous point (4E) should be commented, in a general way, also here, discussing in more detail possible approaches or even prospects on how to replace mercury working electrodes by less-controversial alternatives.
Answer: In Conclusions, a comment about replacing mercury electrodes with more environmentally friendly electrodes has been added.
Remark: 6) Table(s) 1-3; pages 13-20 // Some minor corrections (throughout the data in the tables) …
(A) I recommend the authors to exchange the order of the first and second columns;
(B) Again, replace the highly valent single ions (like V(V) etc.) with the corresponding anions or cationic complexes [like VO3(-) or VO2(+)];
(C) Should be: "chloranilic acid", "…-aldehyde", "ICP-MS" (or "ICP-OES" and not only "ICP", which is the way of ionisation only and not a detection technique / method).
Answer:
(A) In Table 1, the distribution of the first and second columns has been modified to reflect the reviewer's comment.
(B) In Table 1, the highly value single ions have been changed to their corresponding cationic complexes according to the reviewer's comment.
(C) Misspelled names have been changed according to the reviewer's comments at the same time and we thank you for pointing out these errors in this paper.
Yours sincerely,
Prof. Malgorzata Grabarczyk
Round 2
Reviewer 1 Report
-
I have no additional comments.
Author Response
Thank you for your letter and the reviewers’ reports of my paper entitled: “A review on adsorptive stripping voltammetric procedures for the determination of vanadium”. Now I am sending you a revised version of the manuscript, corrected according to the reviewer’s suggestions. The changes made in the revised manuscript are highlighted.
The revisions which we made are as follows:
Reviewer 1:
Remark: I have no additional comments.
Answer: Thank you very much for your positive review of our manuscript.
Yours sincerely,
Prof. Malgorzata Grabarczyk
Reviewer 2 Report
As this study is focused on adsorptive stripping voltammetry, voltammograms and figures from published articles on the same topic must be included, and the authors and publishers must grant permission for republishing. After minor revisions, it may be considered for publication.
Author Response
Thank you for your letter and the reviewers’ reports of my paper entitled: “A review on adsorptive stripping voltammetric procedures for the determination of vanadium”. Now I am sending you a revised version of the manuscript, corrected according to the reviewer’s suggestions. The changes made in the revised manuscript are highlighted.
The revisions which we made are as follows:
Reviewer 2:
Remark: As this study is focused on adsorptive stripping voltammetry, voltammograms and figures from published articles on the same topic must be included, and the authors and publishers must grant permission for republishing. After minor revisions, it may be considered for publication.
Answer: We are extremely sorry but, despite our efforts, we did not obtain permission from the authors and publishers to include voltamperograms and figures already published in publications.
Yours sincerely,
Prof. Malgorzata Grabarczyk
Reviewer 4 Report
Referee's Comments on the Revised Version of:
“A review on adsorptive stripping voltammetric procedures for the determination of vanadium”
(by E. Wlazłowska and M. Grabarczyk; submitted to Materials (MDPI) and registered
as # MATER-23-19500)
In my previous review report, I had evaluated the review paper by E. Wlazłowska and M. Grabarczyk with a conclusion "(it) can be recommended after major revision", having raised six comments with some other sub-notes that should have been considered and, afterwards, adequately reflected in the revised version.
After checking all these corrections in the revised manuscript, I can state that the authors have accepted almost all my comments, incorporating them properly into the new version, which is specified in more detail in my brief commentary below. However, there is still one point that has not been reflected and I insist on doing so. For more details, see below and go into (3B). If the authors, E. Wlazłowska and M. Grabarczyk, reflect satisfactorily this point; then, I will be ready to recommend their manuscript for publication.
Addressed Remarks on the Individual Changes // My evaluation(s)
1) Title // Formal note + Recommendation … Done.
2) Abstract // Critical comment + Correction … (A) Done as I had recommended. (B) After checking it throughout the text, the symbols and formulas were corrected.
3) Introduction // Formal note + Principal critical comment… (A) Done.
(B) Regarding this point, the authors apparently did not understand exactly what I had meant by “the electrochemistry of vanadium and some problems with redox transformations V(V) ® V(IV) ® V(III) ® V(II) and vice versa”.
The authors added the basic principles of adsorptive stripping voltammetry, including a schematic picture, emphasising yet the respective advantages of AdSV mode. But, this is not what I wanted to see here.
I just wanted to briefly explain why the above shown cascade reduction(s) / oxidation(s) are not being used for practical determinations of vanadium. Is it because of poor reaction kinetics ? Or, is there no quantitative character of some of these reactions? Or also, is there any “electroanalytical” reason, such as low sensitivity or poor reproducibility of the corresponding signals ? Or, finally, anything else ?
Please, check again the relevant literature and try to find the answer(s) to my question(s) concerning the specifics of the electrochemistry of vanadium. Then, write newly a short and concise paragraph about it so that the reader could be informed why the individual redox transformations of vanadium cannot be used for direct measurements and the electroanalytical methods must employ the AdSV regime with preconcentration step and subsequent re-oxidation / re-reduction of the products formed during the preconcentration (!)
4) Chapters 2 and 3: AdSV of Vanadium and the WE; pages 2-10, The individual notes … (A-D) Section 2.1 : Done ; (E) Section 2.3 : Done and in a very good way !
5) Applications and Conclusions; pages 11-12 // Comment … Also done.
6) Tables 1-3; pages 13-20 // Minor corrections … (A-C): Done.
Author Response
Thank you for your letter and the reviewers’ reports of my paper entitled: “A review on adsorptive stripping voltammetric procedures for the determination of vanadium”. Now I am sending you a revised version of the manuscript, corrected according to the reviewer’s suggestions. The changes made in the revised manuscript are highlighted.
The revisions which we made are as follows:
Reviewer 4:
Remark: “A review on adsorptive stripping voltammetric procedures for the determination of vanadium”(by E. WlazÅ‚owska and M. Grabarczyk; submitted to Materials (MDPI) and registered as # MATER-23-19500)
In my previous review report, I had evaluated the review paper by E. Wlazłowska and M. Grabarczyk with a conclusion "(it) can be recommended after major revision", having raised six comments with some other sub-notes that should have been considered and, afterwards, adequately reflected in the revised version.
After checking all these corrections in the revised manuscript, I can state that the authors have accepted almost all my comments, incorporating them properly into the new version, which is specified in more detail in my brief commentary below. However, there is still one point that has not been reflected and I insist on doing so. For more details, see below and go into (3B). If the authors, E. Wlazłowska and M. Grabarczyk, reflect satisfactorily this point; then, I will be ready to recommend their manuscript for publication.
3) (B) Regarding this point, the authors apparently did not understand exactly what I had meant by “the electrochemistry of vanadium and some problems with redox transformations V(V) ® V(IV) ® V(III) ® V(II) and vice versa”.
The authors added the basic principles of adsorptive stripping voltammetry, including a schematic picture, emphasising yet the respective advantages of AdSV mode. But, this is not what I wanted to see here.
I just wanted to briefly explain why the above shown cascade reduction(s) / oxidation(s) are not being used for practical determinations of vanadium. Is it because of poor reaction kinetics ? Or, is there no quantitative character of some of these reactions? Or also, is there any “electroanalytical” reason, such as low sensitivity or poor reproducibility of the corresponding signals ? Or, finally, anything else ?
Please, check again the relevant literature and try to find the answer(s) to my question(s) concerning the specifics of the electrochemistry of vanadium. Then, write newly a short and concise paragraph about it so that the reader could be informed why the individual redox transformations of vanadium cannot be used for direct measurements and the electroanalytical methods must employ the AdSV regime with preconcentration step and subsequent re-oxidation / re-reduction of the products formed during the preconcentration (!)
Answer: As suggested by the Reviewer, additional information has been added in the Introduction (page 4, line 137-150).
“In the case of vanadium determination, anodic stripping voltammetry (ASV) is very rarely used, in which the determination is based on redox reactions associated with the change in the oxidation state of vanadium, and few papers on this are available in the literature [41,44]. This is due to the fact that the electrochemical behavior of vanadium is rather complex because of the large number of its oxidation states, its tendency to undergo acid-base reactions, complex formation, and polymerization. Vanadium(V), in particular, forms many species that are strongly pH-dependent. At mercury electrodes, the redox-couple vanadium(V)/(IV) cannot be studied easily in acidic media, because vanadium(V) reacts chemically with mercury, whereas the oxidation of vanadium(IV) does not occur within the attainable potential range [44]. Therefore, the voltammetric determination of vanadium is carried out mainly by the AdSV method. In addition, the AdSV method allows for better sensitivity of determinations, which makes it the first choice method compared to ASV when we want to obtain a low limit of detection.”
Yours sincerely,
Prof. Malgorzata Grabarczyk